# OpenReview forum: "Rethinking the Stability-Plasticity Trade-off in Continual Learning from an Architectural Perspective"
_ICML.cc/2025/Conference — ICML 2025 poster_

### Official Review · Reviewer_2FFZ · 2025-03-04

**Overall Recommendation:** 4

**Summary:**

This paper tackles the problem of offline Class Incremental Learning but leverage a a dual architecture strategy. The goal is to design one architecture that would be more plastic (focus on new knowledge) and another that would be more stable (focus on older knowledge) and combine both capabilities during training through distillation. Notably, the author show via experiments that deep models are more plastic and wider models are more stable. The authors apply such strategy with various state-of-the-arts methods and demonstrate superior performances in two datasets.

## update after rebuttal

The identified limitation of computation is indeed important. Hopefully the authors provided interesting additional experiments demonstrating that the method is still valuable, but is more of a tradeoff than previously presented. Interestingly, the achieved tradeoff is more better than previous method leverage multiple models and similarly, leveraging the same computation gives overall better performances. Eventually, in any cases, such approach is valuable for inference. In that sense, given that the authors include the experiments provided in the rebuttal, and refine the presentation of the method including, making more apparent the drawback between training computation, inference optimization, and memory usage, I recommend accepting the paper. In that sense I raised my score to 4.

**Claims And Evidence:**

- Given the standard deviations in Table 1, the conclusions in 3.2 regarding using a deeper network are not very convincing
- I deeply appreciate that the code is shared
- Other claims are convincing

**Essential References Not Discussed:**

Additionnal methods employing several learners already exist and should be discussed in the current paper. Especially in the case of [2, 3], the authors similarly have a double architecture that focuses on stability and plasticity, so I believe such work should be mentioned.

[2] Michel, Nicolas, et al. "Rethinking Momentum Knowledge Distillation in Online Continual Learning." Forty-first International Conference on Machine Learning.

[3] Lee, Hojoon, et al. "Slow and Steady Wins the Race: Maintaining Plasticity with Hare and Tortoise Networks." International Conference on Machine Learning. PMLR, 2024.

**Experimental Designs Or Analyses:**

See above.

**Methods And Evaluation Criteria:**

- As shown in a previous study [1], the forgetting metric can advantage more plastic model, so it would also be interesting to consider the relative forgetting metric [1]

[1] Wang, Maorong, et al. "Improving plasticity in online continual learning via collaborative learning." Proceedings of the IEEE/CVF Conference on Computer Vision and Pattern Recognition. 2024.

**Other Comments Or Suggestions:**

See above.

**Other Strengths And Weaknesses:**

- The proposed approach leverages fewer parameters but it is unclear in terms of computation since it is not computable in parallel. Similarly, the distillation loss term increases computation. I believe sharing computation time is necessary, in addition to the FLOPs given in the appendix.
- I really like the idea of the paper, however, I have a hard time understanding how the overall training procedure occurs, since both models are optimized. A pseudo-code would really make this procedure clearer in my opinion. In that sense, I checked the code given by the authors, and I am now even more confused. In the code shared, if we check for example the file `models/icarl_t.py`. In the `_init_train()` and `_update_representation()` functions, I can see **two optimization loops of 100 epochs, for each model**. One at line 215 and one at line 248, for `_update_representation()`. This is very important and not mentioned in the paper, from my understanding. Similarly, it impacts the training time. This is for me the main weakness of the paper.

**Questions For Authors:**

- How about model ensemble for inference? Have you tried? I understand that it would increase computation but this might also lead to superior performance.
- Why swapping stab and pla models leads to a strong drop in performances, as observed in the last row of Table 3?
- In section 5.4 the authors claim a reduction in parameters count of 87\% and 81\%, however in the table the reduction is of about 50\% for the considered methods. Could you elaborate on this?
- What does this "Dynamic Arcchitecture" part mean in Figure 2? Is it because the classification head grows during training? I do not find it very clear at the moment.
- Could you give more details on how the hyper-parameters are chosen? Did you conduct any search? Do you use the exact same HP for the baseline and baseline+DualArch?

**Relation To Broader Scientific Literature:**

Key contributions are:
- helping continual learning practitioners design more efficient models in continual learning
- giving hindsight on the architectural design of continual learners

**Theoretical Claims:**

This paper does not contain any theoretical claim.

---

> ### Author Rebuttal · Authors · 2025-04-01
>
> We sincerely thank Reviewer 2FFZ for the recognition of our insightful empirical findings and interesting idea. We are also grateful for the valuable and constructive feedback.
>
> > Conclusions regarding deeper network
>
> Our main conclusion is that "existing architectural designs typically exhibit good plasticity but poor stability, while their wider and shallower variants exhibit the opposite traits" (contribution 1). We acknowledge that further increasing depth further does not significantly boost plasticity, so our plastic network's depth aligns with ResNet-18 (Sec. 4.3). We'll revise Sec. 3.2 to clarify this and avoid ambiguity.
>
> >Relative forgetting
>
> While the Average Forgetting is widely used, we agree that *relative forgetting (RF)* [1] offers a fairer stability measure. We've added RF to our analysis. Results on Tab. 1 (partial) support our original claims.
>
> | Depth-Width | Penu. Layer | RF↓ (%)       |
> | ----------- | ----------- | ------------- |
> | 18-64       | GAP         | 41.09         |
> | 10-96       | GAP         | 39.15 (-1.94) |
> | 18-64       | 4x4 AvgPool | 39.97 (-1.12) |
>
> > Essential References
>
> We thank the reviewer for highlighting these relevant works. We've updated Sec. 2.3 to include a discussion of [2] and [3]. Specifically: (*italicized* indicates new content):
>
> "..., certain works utilize two learners (known as slow and fast learners) with different functions to achieve CL. *Among them, MKD [2] and Hare & Tortoise[3] employ techniques such as exponential moving averages to integrate knowledge across two models, effectively balancing stability and plasticity.* Our proposed solution ..."
>
> > Computation time
>
> *Training*: Despite reduced FLOPs (A.1), training time may increase due to non-parallelism. We note that this common trade-off in multi-model methods (e.g., [1]) is justified by the performance gains. As shown in the below (CIFAR100/10, minutes), our methods require 1.39× to 1.77× time cost for training. We'll include this analysis in this revision.
>
> | Method              | iCaRL | WA   | DER  | Foster | MEMO |
> | --------- | ----- | ---- | ---- | ------ | ---- |
> | Baseline (ResNet18) | 40    | 39   | 74   | 93     | 49   |
> | w/ Ours             | 70    | 69   | 106  | 129    | 85   |
>
> *Inference*: Using only the lightweight main model during inference, our method reduces computational cost, showing a 76% faster speed.
>
> | Models   | Frame per second ↑ |
> | -------- | ------------------ |
> | Baseline | 5452               |
> | w/ Ours  | 9604 (+76%)        |
>
> > Training procedure
>
> As suggested, we'll include pseudo-code to clarify the training and testing procedure. Brief process is as follows:
>
> 1. Initialize the stable main learner $\theta_0$ and plastic auxiliary learner $\phi_0$).
> 2. For each new task t (1 to N):
>    - **Train plastic learner**: Update $\phi\_{t-1}$ for E epochs using normal classification loss to get $\phi_{t}$, then freeze and save it.
>    - **Train stable learner**: Update $\theta_{t-1}$ for E epochs with the plastic learner's assistance and CL method (loss Eq. 1) to get $\theta_t$.
>    - **Test stable learner**: Evaluate $\theta_t$ on all previous tasks (1 to t).
>
> We clarify that our method is **fully transparent**, though some details may need elaboration:
>
> 1. Sequential training of two models is mentioned in Sec. 4.4 (lines 257-258);
>
> 2. Fairness is ensured by comparing total FLOPs of both models against single-model baselines in A.1.
>
> > Model ensemble
>
> We tried ensemble techniques (e.g., exponential moving average) but found them challenging to implement across differing architectures. We believe they hold promise for integrating knowledge and will explore them in our future work.
>
> > Performance of swapping sta and pla networks
>
> The stable and plastic learners have distinct roles (main and auxiliary) in learning, so swapping their architectures degrades performance. This shows the need for dedicated architectures suited to each learner's function.
>
> > Parameters count in Sec. 5.4
>
> "87%" and "81%" refer to Fig. 3, where our method remain effective under stricter parameter constraints.
>
> > Dynamic Architecture in Fig. 2
>
> "Dynamic Architecture" refers to integrating our method with dynamic architecture techniques by expanding the stable learner’s network only. This has been clarified in the revision.
>
> > Hyper-parameters (HP)
>
> For fairness and reproducibility, we use the same HP for both baseline and baseline+DualArch without specific tuning. HPs for CL methods follow the PyCIL library [4], with epochs and optimizer scheduler adjusted for consistency across all methods, as noted in Sec. 5.1.
>
> **References:**
>
> [1] Improving plasticity in online continual learning via collaborative learning. CVPR 2024.
>
> [2] Rethinking Momentum Knowledge Distillation in Online Continual Learning. ICML 2024.
>
> [3] Slow and Steady Wins the Race: Maintaining Plasticity with Hare and Tortoise Networks. ICML 2024.
>
> [4] Pycil: a python toolbox for class-incremental learning. 2023

---

> > ### Comment · Reviewer_2FFZ · 2025-04-02
> >
> > I thank the authors for their detailed feedback and transparency.
> >
> > > We clarify that our method is fully transparent, though some details may need elaboration:
> > >    1.  Sequential training of two models is mentioned in Sec. 4.4 (lines 257-258);
> > >    2. Fairness is ensured by comparing total FLOPs of both models against single-model baselines in A.1.
> >
> > I did miss 1., thank you for the clarification. I agree that 2. is present but as mentioned by myself and other reviewers, computation is more relevant here and is indeed largely impacted by the proposed approach, at least for training. One advantage is the reduced computation for testing I suppose. Another is the reduced memory footprint.
> > I appreciate the honesty to fully include training time. Now I wonder, how does training time of DualArch compare with ArchCraft? Is DualArch more computation-heavy but leverages fewer parameters?
> > Additionally, what happens if the number of **overall epochs** are the same? So instead of having 100 epochs per sub-model (total 200) you have 50 per sub-model (total 100). How does this compare to training a single model 100 epochs?
> >
> > The introduced approach is eventually a tradeoff between memory footprint, training time, and inference speed, which remains to be clearly stated in the current version of the paper.

---

> > > ### Author Response · Authors · 2025-04-02
> > >
> > > We sincerely thank Reviewer 2FFZ for the valuable feedback on our response.
> > >
> > >
> > >
> > > > How does training time of DualArch compare with ArchCraft?
> > >
> > > DualArch exhibits lower training time (minutes) compared to ArchCraft. This demonstrates that DualArch is lightweight in both computation and parameter counts when compared with ArchCraft. We will include these results in the revised manuscript.
> > >
> > > | Method    | iCaRL  | WA     | DER     | Foster  | MEMO   |
> > > | --------- | ------ | ------ | ------- | ------- | ------ |
> > > | ArchCraft | 85     | 86     | 184     | 186     | 107    |
> > > | w/ Ours   | **70** | **69** | **106** | **129** | **85** |
> > >
> > >
> > >
> > > > What happens if the number of **overall epochs** are the same?
> > >
> > > As suggested, we conducted an ablation study where we **halved the training epochs** of DualArch to match the total epochs of the baselines (on CIFAR100/10). The results show that DualArch with reduced epochs achieves comparable performance, while reducing training times for most of methods. We appreciate this insightful comment and will include these findings in the revision.
> > >
> > > **Results of LA % ↑:**
> > >
> > > | Method                    | iCaRL     | WA        | DER       | Foster    | MEMO      |
> > > | ------------------------- | --------- | --------- | --------- | --------- | --------- |
> > > | Baseline (ResNet18)       | 54.87     | **56.98** | 61.83     | 58.67     | 58.46     |
> > > | w/ Ours + **Half epochs** | **57.00** | 55.65     | **63.83** | **61.84** | **64.17** |
> > >
> > > **Results of AIA % ↑:**
> > >
> > > | Method             | iCaRL     | WA        | DER       | Foster    | MEMO      |
> > > | ------------------------- | --------- | --------- | --------- | --------- | --------- |
> > > | Baseline (ResNet18)       | 68.30     | **69.16** | **72.48** | 69.95     | 70.71     |
> > > | w/ Ours + **Half epochs** | **68.31** | 65.56     | 71.42     | **71.60** | **71.09** |
> > >
> > > **Training time (minutes) ↓:**
> > >
> > > | Method            | iCaRL  | WA     | DER    | Foster | MEMO   |
> > > | ------------------------- | ------ | ------ | ------ | ------ | ------ |
> > > | Baseline (ResNet18)       | 40     | 39     | 74     | 93     | **49** |
> > > | w/ Ours + **Half epochs** | **36** | **35** | **55** | **63** | 53     |
> > >
> > >
> > >
> > > > The introduced approach is eventually a tradeoff between memory footprint, training time, and inference speed, which remains to be clearly stated in the current version of the paper.
> > >
> > > As suggested, we will incorporate a detailed discussion of the tradeoffs between memory footprint, training time, and inference speed in this revision. This will include an analysis of both the advantages and limitations of our approach, highlighting the key gains as well as potential challenges.
> > >
> > >
> > >
> > > We thank Reviewer 2FFZ for the constructive comments again, which have significantly enhanced the clarity and overall quality of our paper.

---

### Official Review · Reviewer_Kz16 · 2025-03-11

**Overall Recommendation:** 2

**Summary:**

This paper studies the stability-plasticity trade-off in continual learning from an architectural perspective. It finds that increasing depth improves plasticity, while increasing width enhances stability. Motivated by this, it proposes a dual-architecture framework, DualArch, comprising two distinct networks dedicated to plasticity and stability, respectively. The proposed framework can be incorporated into existing continual learning methods to enhance performance while remaining significantly parameter efficient.

## update after rebuttal
After the rebuttal, my primary concerns regarding scalability and computational efficiency remain unaddressed. The experimental validation continues to rely on limited-scale settings, with the ImageNet-1K results offering minimal insight due to significant downsampling. Additionally, the reported increase in training time relative to baselines, even on smaller datasets like CIFAR-10/100, raises further concerns about the method’s practicality in real-world, large-scale scenarios. Therefore, my assessment of the paper remains unchanged.

**Claims And Evidence:**

The paper provides evidence supporting the efficacy of DualArch in small-scale continual learning settings. It demonstrates that DualArch enhances both plasticity and stability, thereby improving overall performance. However, I have the following concerns:

1. **Compute Overhead**: While DualArch claims to be parameter efficient, it is unclear how it enhances computational efficiency given it involves training two models in two distinct training stages. This setup seemingly incurs approximately 2x compute overhead compared to training a single network. A direct comparison in terms of compute cost (e.g., number of updates or FLOPs) would provide more insight.

2. **Scalability**: The proposed framework is only evaluated on datasets with a maximum of 100 classes. It remains unclear how DualArch would perform on larger datasets like ImageNet-1K or longer task sequences.

**Essential References Not Discussed:**

The paper seems to discuss relevant prior work adequately.

**Experimental Designs Or Analyses:**

The experiments appear reasonable but do not study *scalability* to larger datasets or tasks. Given the proposed method's reliance on training two models, computational overhead could significantly increase in large-scale tasks.

**Methods And Evaluation Criteria:**

The forgetting metric (as presented in Figure 1a and tables) appears problematic since it is inherently model-dependent. A robust forgetting evaluation typically compares the model's performance against a universal upper bound (a jointly trained model on the entire dataset/sequence). Without this upper bound, the forgetting metric lacks interpretability.

For example:
- **Model 1**: Learns less, forgets less → Final accuracy: $x$
- **Model 2**: Learns more, forgets more → Final accuracy: $y$
- **Joint model**: Upper bound accuracy: $z$

In some cases, $z - x > z - y$, meaning that Model 2 actually learns and retains more knowledge despite higher forgetting. Hence, the current forgetting metric can be misleading.

Additionally, the paper does not report computational efficiency (e.g., FLOPs, training time) compared to a single-model baseline or training-from-scratch models (joint models). This is critical for real-world continual learning applications where a continual learner should be more efficient than the jointly trained model.

**Other Comments Or Suggestions:**

Please see my comments above regarding weaknesses.

**Other Strengths And Weaknesses:**

Striking a better stability-plasticity trade-off from an architectural perspective seems intriguing. The idea presented in the paper seems interesting, but performing more rigorous experiments will make the paper stronger. In particular, compute efficiency and ability to scale are important criteria to align continual learning with real-world applications.

**Questions For Authors:**

1. How does DualArch perform on datasets with more than 100 classes, such as ImageNet-1K or subsets?
2. Are the plastic and stable networks initialized with random or pre-trained weights?
3. Can you provide a comparison of the compute cost (in FLOPs or training time) compared to training a single model?

**Relation To Broader Scientific Literature:**

While prior studies separately identify that wider network favors stability or deeper one enhances plasticity, there exists a gap to unify these findings. This paper unifies stability and plasticity from an architectural perspective, which may be informative for the research community.

**Theoretical Claims:**

Not applicable

---

> ### Author Rebuttal · Authors · 2025-04-01
>
> We sincerely thank Reviewer Kz16 for the recognition of our intriguing research perspective and interesting idea. We are also grateful for the valuable and constructive feedback.
>
> > Computation overhead.
>
> We would like to clarify that the total FLOPs of two models (Sta-Net and Pla-Net) in Dual-Arch is lower than the baseline, as detailed in Appendix A.1 (lines 614-617). For instance, on CIFAR100, Sta-Net and Pla-Net require 255M and 241M FLOPs, respectively, totaling 496M—less than the 558M FLOPs of ResNet-18. We thank the reviewer for the valuable suggestion and will emphasize this point in the revised main text for clarity.
>
> *Training time*: While the total FLOPs are reduced as shown in A.1, we acknowledge that the actual training time may increase due to non-parallel training. We note that this trade-off is common in multi-model methods (e.g., [1]) and is justified by the performance gains. As shown in the below CIFAR100/10 experiments, our methods require 1.39× to 1.77× time cost for training.  We will include this analysis in this revision.
>
> | Method              | iCaRL  | WA     | DER     | Foster  | MEMO   |
> | ------------------- | ------ | ------ | ------- | ------- | ------ |
> | Baseline (ResNet18) | 40 min | 39 min | 74 min  | 93 min  | 49 min |
> | w/ Ours             | 70 min | 69 min | 106 min | 129 min | 85 min |
>
> *Inference time*: Since our approach utilizes only the main model (a stable learner) during inference, which is more lightweight compared to the baseline ResNet-18, the computational cost during inference is significantly reduced. Below are the results on ImageNet, demonstrating that our method achieves a 76% increase in inference speed.
>
> | Models              | Frame per second ↑ |
> | ------------------- | ------------------ |
> | Baseline (ResNet18) | 5452               |
> | w/ Ours             | 9604 (+76%)        |
>
> > Long task sequences and large datasets.
>
> *Large datasets:* CIFAR100 and ImageNet100 are common benchmarks in continual learning (CL) research[1-3]. To further address the reviewer's concern, we extend our evaluation to large-scale ImageNet-1K/10 tasks. Following [4], we use the 32×32 downsampled ImageNet-1K for efficiency, with a replay buffer size of 20,000 as in [5]. Results show our method remains effective, achieving gains of +3.37% and 5.11%, indicating its strong scalability.
>
> | Methods | AIA /%            |
> | ------- | ----------------- |
> | iCaRL   | 24.25             |
> | w/ Ours | **27.62 (+3.37)** |
> | WA      | 22.42             |
> | w/ Ours | **27.53 (+5.11)** |
>
> *Long task sequences:* Tab. 7 (Appendix A.5) demonstrates our method’s superior performance on CIFAR100/50, a benchmark with long task sequences (50 tasks). The results below highlight consistent improvements of our method over baselines.
>
> | Method  | LA (%)    | AIA (%)   |
> | ------- | --------- | --------- |
> | iCaRL   | 45.30     | 63.99     |
> | w/ Ours | **48.95** | **65.93** |
> | WA      | 42.12     | 58.26     |
> | w/ Ours | **47.13** | **64.41** |
>
> > Forgetting metrics.
>
> We would like to clarify that (final) average forgetting (FAF) is a well-established metric in continual learning for measuring stability [4, 5]. Regarding the reviewer's suggestion to compare the model's performance against a universal upper bound, we respectfully note that such an evaluation may represent overall performance rather than forgetting.  Since the upper bound is shared across methods, the resulting metric would primarily reflect final/last accuracy (LA), which inherently combines both plasticity (AAN) and forgetting (FAF), as approximated by $LA \gtrapprox AAN - FAF$.
>
> > Initialization of networks.
>
> All models are randomly initialized, consistent with baseline methods. We will add a pseudo-code in this revision to clarify this point. Here is a brief process description:
>
> 1. Randomly initialize the stable learner (main model, $\theta_0$) and plastic learner (auxiliary model, $\phi_0$).
> 2. For each new task t (1 to N):
>    - **Train plastic learner**: Update $\phi\_{t-1}$ for E epochs using normal classification loss to get $\phi_{t}$, then freeze and save it.
>    - **Train stable learner**: Update $\theta_{t-1}$ for E epochs with the plastic learner's assistance and CL method (loss Eq. 1) to get $\theta_t$.
>    - **Test stable learner**: Evaluate $\theta_t$ on all previous tasks (1 to t).
>
> **References:**
>
> [1] Rethinking Momentum Knowledge Distillation in Online Continual Learning. ICML 2024.
>
> [2] Resurrecting old classes with new data for exemplar-free continual learning. CVPR 2024.
>
> [3] Incorporating neuro-inspired adaptability for continual learning in artificial intelligence. Nature Machine Intelligence 2023
>
> [4] Loss of plasticity in deep continual learning. *Nature* 2024.
>
> [5] Pycil: a python toolbox for class-incremental learning. 2023.
>
> [6]  A comprehensive survey of continual learning: Theory, method and application.  IEEE TPAMI 2024.
>
> [7] Class-incremental learning: A survey. IEEE TPAMI 2024.

---

> > ### Comment · Reviewer_Kz16 · 2025-04-05
> >
> > Thank you for your response. I will maintain my original score.
> >
> > **Additional Comments**
> >
> > I'm sorry for not clarifying my concerns. Previously, I posted my additional comments using "official comment", which was not visible to the authors. I have just realized that. Hence, I am adding my additional comments here.
> >
> > My primary concerns revolve around scalability and efficiency, which are critical for real-world applications. While toy settings can be useful for demonstrating proof of concept, they provide limited insight into whether a method will be effective in practical scenarios involving a large number of data points, a higher number of classes, high-resolution images, and longer task sequences. Unfortunately, these concerns remain unaddressed in the rebuttal. The ImageNet setting presented is not ideal, as it fails to incorporate the aforementioned criteria. Specifically, downsampling images to $32 \times 32$ significantly limits the evaluation of scalability on high-resolution datasets (e.g., $224 \times 224$ images for ImageNet).
> >
> > Furthermore, compute and memory overhead are crucial factors in large-scale settings, and datasets like CIFAR-100 are insufficient for evaluating such constraints. In CIFAR-10/100 experiments, the proposed method requires $1.39\times$ to $1.77\times$ the training time compared to the baselines. This raises further concerns about its computational efficiency in large-scale scenarios.
> >
> > Ultimately, it is vital to carefully assess whether findings from toy settings will generalize, as many methods that perform well on simplified benchmarks often fail to scale to more complex, real-world datasets.

---

> > > ### Author Response · Authors · 2025-04-06
> > >
> > > We thank the reviewer for the response. We sincerely appreciate the reviewer's time and effort in reviewing our work and have incorporated all suggestions into this revision, including:
> > >
> > > - Clarifying computational overhead comparisons (FLOPs, training/inference time) to address the concern about efficiency.
> > > - Extending evaluations to ImageNet-1K as suggested for large-scale validation.
> > > - Explicitly stating network initialization details.
> > >
> > > We believe these revisions have strengthened the manuscript and fully address the reviewer's concerns. We remain open to further suggestions to improve the work.

---

### Official Review · Reviewer_MGyK · 2025-03-13

**Overall Recommendation:** 5

**Summary:**

The paper investigates the stability-plasticity trade-off in continual learning from an architectural perspective. Through empirical studies, the authors find that depth enhances plasticity while width favors stability. Building on this insight, they propose an approach that leverages two specialized networks with complementary strengths (stability and plasticity) to address this trade-off. Extensive experiments across multiple datasets and continual learning methods demonstrate that the proposed method not only improves performance but also enhances parameter efficiency compared to baselines.

## update after rebuttal

After the rebuttal, all of my concerns have been adequately addressed. I have also read other reviewers' comments. I think the experimental validation is already sufficient, based on my knowledge of the continual learning. Regarding training time, I also agree with Reviewer 2FFZ's updated assessment that it is acceptable. For these reasons, I tend to accept this submission.

**Claims And Evidence:**

The authors conduct thorough empirical studies comparing different architectural configurations (depth vs. width)  in Section 3, providing clear evidence supporting their claims regarding the trade-off between stability and plasticity at the architectural level. The ablation study also effectively support their claims.

**Essential References Not Discussed:**

The related works section is comprehensive, covering both learning methods and neural architectures for continual learning. To the best of my knowledge, no key related works are missing.

**Experimental Designs Or Analyses:**

I have examined the implementation details and reviewed the provided source code. The proposed method do achieve a fair improvement compared with the baselines.

**Methods And Evaluation Criteria:**

The proposed method presents a simple yet effective solution to the stability-plasticity dilemma by combining the strengths of two specialized architectures. The evaluation criteria, including benchmark datasets and metrics, are consistent with established conventions and well-suited for validating the proposed method.

**Other Comments Or Suggestions:**

- In line 113, "exploit it" should be corrected to "exploited."
- There is inconsistent formatting with "w/ Ours" and "w/ ours."
- The font size in Figure 3 is too small.

**Other Strengths And Weaknesses:**

*Strengths:*
1. The paper provides a novel and insightful perspective on continual learning by highlighting the inherent conflict between stability and plasticity at the architectural level. The empirical finding that depth benefits plasticity while width enhances stability has the potential to inspire future research to enhance continual learning by exploring more balanced architectures.
2. After reading the line 86-107, I found the proposed method is simple yet intuitive. Moreover, its effectiveness in combining the strengths of two specialized architectures is well validated through experiments and analysis.
3. The performance results presented in Table 2 are impressive. The method demonstrates significant and consistent improvements across different continual learning methods and benchmarks, while also achieving substantial gains in parameter efficiency.
4. The paper is well-written, clearly structured, and easy to follow.

*Weaknesses:*
1. Recent studies have increasingly focused on replay-free settings. Despite that the effectiveness of the proposed method has been well-demonstrated across various continual learning approaches, further validation in replay-free settings is still beneficial.
2. The discussion of "Architecture-based methods" in the related work section may be somewhat confusing. I recommend the authors clarify why these methods belong under "*Learning Methods for CL*" rather than "*Neural Architectures for CL*."
3. The rules for computing parameter counts are reasonable but differ from those used in certain existing studies. To avoid confusion, I recommend the authors detail these rules in the main paper.

**Questions For Authors:**

Could the authors provide an analysis of the proposed method's effectiveness when combined with state-of-the-art replay-free continual learning methods?

**Relation To Broader Scientific Literature:**

While existing research focuses on optimizing weights to address the stability-plasticity trade-off at the parameter level, this study introduces a novel perspective by extending this trade-off to the architectural level. Moreover, by exploring and addressing the inherent conflict between stability and plasticity at the architectural level, the paper distinguishes itself from existing exploration on neural architectures for continual learning.

**Theoretical Claims:**

This paper is primarily motivated by empirical studies, so there are no formal theoretical proofs to verify.

---

> ### Author Rebuttal · Authors · 2025-04-01
>
> We sincerely thank Reviewer MGyK for the recognition of our insightful research perspective, simple yet effective method, and impressive performance improvements. We are also grateful for the valuable and constructive feedback.
>
> > Validation on replay-free settings.
>
> While our current experiments primarily focus on replay-based continual learning settings, our approach is also compatible with state-of-the-art replay-free methods. To demonstrate this, we evaluate our approach in conjunction with three representative replay-free continual learning methods: LWF [1], ADC [2], and LDC [3] as representative replay-free methods. The results (LA /%) consistently show that our Dual-Arch approach enhances the performance of all baseline methods, underscoring its versatility and effectiveness in replay-free continual learning scenarios.
>
> | Method  | CIFAR100/20 | CIFAR100/10 |
> | ------- | ----------- | ----------- |
> | LWF     | 16.98       | 26.63       |
> | w/ Ours | **20.27**   | **33.81**   |
> | ADC     | 30.48       | 40.25       |
> | w/ Ours | **31.90**   | **40.69**   |
> | LDC     | 31.61       | 41.57       |
> | w/ Ours | **34.69**   | **42.91**   |
>
> > Architecture-based methods.
>
> We appreciate the reviewer’s valuable comments and will revise Sections 2.1 and 2.2 to clarify the distinction between two key concepts. To clarify, architecture-based methods primarily focus on dynamically expanding or allocating networks for each task based on a given basic architecture. In contrast, "Neural Architecture for CL" refers to studies that explore and optimize the basic architecture itself to better suit continual learning objectives.
>
> > Parameter counts rules.
>
> We thank the reviewer for raising this point. To avoid any confusion, we will update the description to explicitly state that the reported parameter counts include all used networks (not just the main network).
>
> > Typo issues and Figure presentation.
>
> We sincerely appreciate the reviewer’s attention to detail. All mentioned typos have been corrected, and we have increased the font size in Figure 3 for better readability. We will thoroughly proofread the entire paper to ensure clarity and correctness.
>
> **References:**
>
> [1] Li, Zhizhong, and Derek Hoiem. Learning without forgetting. IEEE TPAMI 2017.
>
> [2] Goswami, Dipam, et al. Resurrecting old classes with new data for exemplar-free continual learning. CVPR 2024.
>
> [3] Gomez-Villa, Alex, et al. Exemplar-free continual representation learning via learnable drift compensation. ECCV 2024.

---

### Official Review · Reviewer_xKgH · 2025-03-13

**Overall Recommendation:** 3

**Summary:**

This paper investigates the stability-plasticity trade-off in continual learning (CL) from an architectural perspective. The authors empirically demonstrate that deeper networks favor plasticity, while wider networks enhance stability under fixed parameter constraints. To address this, they propose Dual-Arch, a framework combining two specialized architectures: a wide/shallow "Sta-Net" for stability and a deep/narrow "Pla-Net" for plasticity.

**Claims And Evidence:**

**Generalizability of architectural insights:**
Experiments are limited to ResNet variants and small datasets. ViT/MLP results (Tables 4–6) show inconsistent trends (e.g., deeper MLPs improve AAN but harm FAF), weakening the claim.

**Essential References Not Discussed:**

No.

**Experimental Designs Or Analyses:**

**Architectural trade-off as primary performance driver:**
Ablation results (Table 3) show that adding any second model (even non-specialized architectures) improves performance, suggesting gains may stem from dual-model integration rather than architectural specialization. For example, using two Sta-Nets (row 4, Table 3) achieves 72.27% AIA vs. 72.92% for Dual-Arch—only a 0.65% gap. If I understand correctly, in your method design, the auxiliary model is re-initialized as new tasks appear. Then the plasticity of an initialized model is stronger than that of a model that has been trained on a certain task [1]. This can also explain why the algorithm has better performance, because the plasticity of the entire system is improved, but this still has no direct connection with your core proposition.

[1] Dohare S, Hernandez-Garcia J F, Lan Q, et al. Loss of plasticity in deep continual learning[J]. Nature, 2024, 632(8026): 768-774.

**Methods And Evaluation Criteria:**

**Dataset limitations**: Experiments on CIFAR100/ImageNet-100 may not reflect real-world scalability. No validation on larger datasets (e.g., ImageNet-1K).

**Other Comments Or Suggestions:**

1. Can you clarify why iCaRL was chosen over simpler baselines for architectural analysis?

2. Can you provide a theoretical explanation (e.g., via NTK or feature learning) for why depth/width affect stability/plasticity?

**Other Strengths And Weaknesses:**

No

**Questions For Authors:**

No

**Relation To Broader Scientific Literature:**

N/A

**Theoretical Claims:**

No, because this paper does not include formal theoretical proofs. The claims are primarily empirical.

---

> ### Author Rebuttal · Authors · 2025-04-01
>
> We sincerely thank Reviewer xKgH for the valuable and constructive feedback.
>
>  > Generalizability of architectural insight / Concern about supplementary material.
>
> There may be a misunderstanding regarding Tab. 5 and 6 results:
>
> - **Shallower yet wider** ViT (5×49, Tab. 5): Lower AAN/FAF value → **reduced plasticity, improved stability** .
> - **Shallower yet wider** MLP (3×1050, Tab. 6): Lower AAN/FAF value → **reduced plasticity, improved stability** .
> - **Deeper yet narrower** MLP (5×680, Tab. 6): Higher AAN/FAF value → **improved  plasticity, reduced stability** .
>
> These align with ResNet trends (Table 1), supporting our architectural insights (depth -> plasticity, width -> stability).
>
> > Performance gains in Tab. 4.
>
> We believe that the performance gains on ViT (minimum +2.47%, maximum +5.88%) is not very modest, especially considering the fact that our method reduces 30% parameter counts.
>
> > Concern about dataset limitations.
>
> CIFAR100 and ImageNet100 are common benchmarks in continual learning (CL) research[1-3]. To further address the reviewer's concern, we extend our evaluation to large-scale ImageNet-1K/10 tasks. Following [4], we use the 32×32 downsampled ImageNet-1K for efficiency, with a replay buffer size of 20,000 as in [5]. Results show our method remains effective, achieving gains of +3.37% and 5.11%, indicating its strong scalability.
>
> | Methods | AIA /% ↑          |
> | ------- | ----------------- |
> | iCaRL   | 24.25             |
> | w/ Ours | **27.62 (+3.37)** |
> | WA      | 22.42             |
> | w/ Ours | **27.53 (+5.11)** |
>
> > Concern about primary performance driver.
>
> We clarify two potential misunderstandings regarding the performance drivers of Dual-Arch.
>
> a) The auxiliary (plastic) model’s weights ($\phi_t$) are **not reinitialized** but are **updated incrementally** from the previous task’s weights ($\phi_{t-1}$), as shown in Fig. 2. So the performance gains could **not** stem from the plasticity of a freshly initialized model. We will further clarify this in Sec. 4.4 of this revision.
>
> b) Dual-Arch's efficacys stems from two key components as mentioned in Sec. 5.3 (lines 324-325):
>
> 1. Dual networks (stable and plastic learner)
> 2. Dedicated architectures (Sta-Net as stable learner, Pla-Net as plastic learner)
>
> While dual Sta-Nets perform strongly (still worse than Dual-Arch), this does not weaken our core proposition. Instead, it highlights the importance of:
>
> - Dual networks (dual Sta-Nets still involve stable and plastic learner)
> - Dedicated architectures (Sta-Net as the stable learner)
>
> To further validate the importance of dedicated architectures, we compare against dual ResNet-18 networks. Results (AIA %) show that our specialized architectures achieve better performance with fewer parameters.
>
> | Architectures          | Parameters (iCaRL) | Avg on Five methods |
> | ---------------------- | ------------------ | ------------------- |
> | Dual ResNet-18         | 33.6M              | 71.46               |
> | Ours (Sta-Net+Pla-Net) | 15.1M (↓55%)       | 72.94 (+1.48)       |
>
> > Why iCaRL was chosen?
>
> Simpler baselines like Experience Replay (ER) may not fully suit CL evaluation—particularly stability. A key limitation is that without additional CL strategies, performance on old tasks might not primarily rely on model's knowledge preservation.
>
> To investigate this, we introduce an ablation on CIFAR100/10: "w/ buffer only", where models:
>
> - Train **only on the last five tasks**,
> - Use an initial replay buffer from the first five tasks,
> - Evaluate performance on the first five tasks after learning the final task.
>
> | Method         | Avg on Task1-5 |
> | -------------- | -------------- |
> | ER             | 38.48          |
> | w/ buffer only | 36.38 (-2.10)  |
> | iCaRL          | 48.64          |
> | w/ buffer only | 41.40 (-7.24)  |
>
> Results show the ablation achieves relatively comparable performance on tasks 1–5 to original ER, despite no direct learning. In contrast, iCaRL shows a larger gap (7.24% vs. 2.10%), indicating the model preserves knowledge beyond buffer memorization. This highlights iCaRL’s suitability for evaluating CL architectures beyond ER.
>
> > Theoretical explanation.
>
> While theoretical analysis (e.g., NTK or feature learning) could offer deeper insights, it is beyond this work’s scope. Existing theories often rely on idealized assumptions (e.g., infinite/very large width or two-layer networks), making extensions to our empirical width/depth trade-offs highly non-trivial.
>
> **References**:
>
> [1] Rethinking Momentum Knowledge Distillation in Online Continual Learning. ICML 2024.
>
> [2] Resurrecting old classes with new data for exemplar-free continual learning. CVPR 2024.
>
> [3] Incorporating neuro-inspired adaptability for continual learning in artificial intelligence. Nature Machine Intelligence 2023.
>
> [4] Loss of plasticity in deep continual learning. *Nature* 2024.
>
> [5] Pycil: a python toolbox for class-incremental learning. 2023.

---

### Decision · Program_Chairs · 2025-05-01

**Decision:**

Accept (poster)

**Comment:**

The authors do a careful empirical study of the importance of width and depth in terms of stability and plasticity. Based on these insights they propose a dual architecture approach, which employs a shallow model for plasticity and a deep one for stability. All reviewers agreed that the work is of good quality and there is sufficient empirical evidence for the observation and proposed method. In the rebuttal the authors provide additional results that solidify their contribution by looking at larger scale models. Overall I recommend acceptance of this work.